# Modelling Metabolic Shifts during Cardiomyocyte Differentiation, Iron Deficiency and Transferrin Rescue Using Human Pluripotent Stem Cells

**DOI:** 10.3390/metabo12010009

**Published:** 2021-12-22

**Authors:** Benjamin B. Johnson, Johannes Reinhold, Terri L. Holmes, Jamie A. Moore, Verity Cowell, Andreia S. Bernardo, Stuart A. Rushworth, Vassilios Vassiliou, James G. W. Smith

**Affiliations:** 1Faculty of Medicine and Health Sciences, Norwich Medical School, University of East Anglia, Norwich NR4 7UQ, UK; Benjamin.B.Johnson@uea.ac.uk (B.B.J.); Terri.Holmes@uea.ac.uk (T.L.H.); Jamie.Moore@uea.ac.uk (J.A.M.); V.Cowell@uea.ac.uk (V.C.); S.Rushworth@uea.ac.uk (S.A.R.); V.Vassiliou@uea.ac.uk (V.V.); 2Developmental Biology Laboratory, Francis Crick Institute, London NW1 1AT, UK; Andreia.Bernardo@crick.ac.uk; 3National Heart and Lung Institute, Imperial College London, London SW3 6LY, UK

**Keywords:** iron deficiency, cardiomyocytes, pluripotent stem cells

## Abstract

Cardiomyocytes rely on specialised metabolism to meet the high energy demand of the heart. During heart development, metabolism matures and shifts from the predominant utilisation of glycolysis and glutamine oxidation towards lactate and fatty acid oxidation. Iron deficiency (ID) leads to cellular metabolism perturbations. However, the exact alterations in substrate metabolism during ID are poorly defined. Using human induced pluripotent stem cell-derived cardiomyocytes (hiPSC-CM), the present study investigated changes in major metabolic substrate utilisation in the context of ID or upon transferrin rescue. Typically, during hiPSC-CM differentiation, the greatest increase in total metabolic output and rate was seen in fatty acid metabolism. When ID was induced, hiPSC-CMs displayed increased reliance on glycolytic metabolism, and six TCA cycle, five amino acid, and four fatty acid substrates were significantly impaired. Transferrin rescue was able to improve TCA cycle substrate metabolism, but the amino acid and fatty acid metabolism remained perturbed. Replenishing iron stores partially reverses the adverse metabolic changes that occur during ID. Understanding the changes in metabolic substrate utilisation and their modification may provide potential for discovery of new biomarkers and therapeutic targets in cardiovascular diseases.

## 1. Introduction

The heart is an organ of perpetually high energy demand. Its specialised cellular processes, including ion transport, sarcomere function, and intracellular Ca^2+^ homeostasis, demand a consistent high level of ATP production [1]. To meet this demand, cardiomyocytes rely heavily on mitochondrial oxidative phosphorylation, and their mitochondria form a dense and complex network organised around the myofibers, which occupies ~30% of the cellular volume, and produces 90% of the required cellular ATP [2]. During heart failure, mitochondrial structure and oxidative function become impaired, decreasing ATP production and resulting in reduced cardiac function [3,4,5,6,7]. Within the failing heart environment, the metabolism of energy substrates shifts in compensatory mechanisms [8,9,10]. Targeting these changes, and altering the relative contribution of different substrates to mitochondrial ATP production, is emerging as a novel therapeutic approach to heart failure [11].

Human induced pluripotent stem cells (hiPSCs) can be differentiated into cardiomyocytes (hiPSC-CMs) to model cardiac diseases and successfully recapitulate abnormal metabolic phenotypes [12,13,14,15]. In the undifferentiated state, hiPSCs exhibit immature mitochondria with low cristae density, relying mostly on glycolysis and glutamine oxidation metabolism [16]. During cardiomyocyte differentiation and maturation in culture, mitochondria elongate and increase their membrane potential, resulting in hiPSC-CMs with higher mitochondrial content, that rely mostly on lactate and fatty acid oxidative metabolism [16]. The metabolic substrates available in the media during differentiation/maturation can have a significant influence on the cellular phenotype produced. Studies have shown that culturing hiPSC-CMs in fatty acid-containing medium enhances the maturation process [17], whilst lactate acid-containing medium can drive opposing de-differentiation [18]. This suggests that the relative shift form glycolysis to oxidative phosphorylation may be a cause, rather than a consequence, of the phenotypic alterations of the hiPSC-CM maturation process. This highlights the importance of specific substrate metabolism on controlling cardiac function.

Iron is an important micronutrient, essential for mitochondrial function [19] and cardiomyocyte metabolism [20]. Iron sulphur (FeS) clusters are ancient and highly preserved structures that are versatile enzymatic co-factors in numerous metabolic pathways [21]. For example, the mammalian complex I (NADH: ubiquinone oxidoreductase) of the electron transfer chain, one of the largest and well-characterised protein complexes in the cell [22], contains eight essential FeS clusters. Complex I acts as the entry-point of electrons into the electron transfer chain (ETC) and is therefore a key enzyme for aerobic metabolism. Mechanistically, after NADH oxidation by the flavin mononucleotide site, electrons are transferred through a chain of seven FeS clusters and used to reduce ubiquinone to ubiquinol at the inner mitochondrial membrane. Loss of FeS clusters due to defective biogenesis or iron deficiency causes metabolic reprogramming with citrate accumulation and cytosolic lipid drop formation [23]. In the heart, loss of NDUSF4, an FeS cluster-containing subunit of complex I, causes reduced complex I activity, cardiac dysfunction and left ventricular hypertrophy [24]. However, the exact metabolic changes that occur in cardiomyocytes during iron deficiency have not been elucidated yet.

Iron deficiency (ID), with or without anaemia, has been identified as an important predictor of adverse outcomes in patients with heart failure, and intravenous replacement has been shown to improve prognosis [25,26,27,28]. Models, both in vivo [29] and in vitro [30], have shown that ID induces a foetal-like glycolytic shift in cardiac metabolism, indicating a potential mechanism for the detrimental consequences of ID in heart failure patients. Yet, the alterations in substrate metabolism during ID are poorly defined, as are the alterations that occur during iron replacement therapies.

In this study, we use a novel, dynamic 96-well microplate assay that is based on colorimetric changes representing electron fluxes through the mitochondrial ETC. This assay assesses a panel of 31 different substrates that are provided to cells that have been permeabilised to ensure uptake. These substrates, utilised in various metabolic pathways and facilitated by specific enzymes and co-factors, provide either nicotinamide adenine dinucleotide (NADH) or flavin adenine dinucleotide (FADH2), ultimately supplying electrons to complex I or complex II, respectively. Electrons then transition through the ETC and are transferred to a tetrazolium redox dye which acts as a terminal electron acceptor turning purple during the reduction [31]. This assay has recently emerged as a simple and reproducible tool for the assessment of mitochondrial metabolism thereby advancing our knowledge of various diseases and pathologies [32,33,34,35,36].

The present study aimed to determine how substrate metabolism, utilised within glycolysis, fatty acid, amino acid and TCA pathways, altered during hiPSC-CM differentiation, and in media containing or devoid of iron. We were able to show that ID leads to impaired metabolism of TCA cycle, amino acid, and fatty acid substrates, whereas glycolytic metabolism was maintained. Transferrin was unable to rescue the phenotype completely. Thus, by investigating the rates and maximal levels of substrate metabolism in detail, we were able to show that hiPSC-CMs can be used to monitor substrate shifts during disease and upon treatment. Having a metabolic insight will allow for better and tailored treatments for specific cardiac metabolic disorders.

## 2. Results

### 2.1. hiPSC-CMs Undergo Critical Metabolic Changes between Day 10 and Day 20 of Differentiation

To ensure the suitability of hiPSC-CMs to model a metabolic substrate shift in ID we first monitored this shift during differentiation. We performed a mitochondrial function assay (Biolog MitoPlate assay) along hiPSC-CM differentiation, to measure the rate of metabolism of a set of 31 substrates, grouped into glycolysis, TCA cycle, amino acid, or fatty acid substrates (Figure 1A). Mitochondrial function was assayed by measuring the rates of dye reduction from electrons flowing into and through the electron transport chain from substrates whose oxidation produces NADH or FADH2. At the early stages of hiPSC-CM differentiation (D6 and D10) the maximal metabolic level of all substrates was decreased (averaging a 30.5% reduction) compared to undifferentiated D0 hiPSCs (Figure 1A). These maximal metabolic levels increased for all substrates at later stages of differentiation (D20 and D30). In both the pluripotent D0 stage and the cardiomyocyte D30 stage, TCA substrate metabolism accounted for the greatest average metabolic output, contributing 104.9 and 220.6 AU, respectively (Figure 1B). Of the TCA cycle substrates, the greatest fold increase from D0 to D30 was seen in the metabolism of aconitic acid (3.8 fold), catalysed by aconitases containing iron-sulphur clusters [37]. The greatest increase in average metabolic output was seen in fatty acid substrate metabolism, increasing from 50.5 AU at D0 to 137.2 AU at D30. Despite the increased fatty acid substrate metabolism observed at D30, there were no significant changes in the relative abundance of the mitochondrial electron transfer chain (ETC) protein complexes (Figure 1C). However, we observed that the metabolism of carnitine, a key fatty acid shown to promote maximal cellular fatty acid uptake and content in hiPSC-CMs [38], was significantly increased at D20, with no further increase observed at D30 (Figure 1D). Moreover, D30 hiPSC-CMs were capable of metabolising short-(acetyl-l-), medium-(octanoyl-l-), and long-(palmitoyl-dl-) chain carnitine forms to similar levels. In all, these results show that there is a marked metabolic shift between D10 and D20 hiPSC-CMs, but that D20 and D30 hiPSC-CMs display a similar metabolism, whereby the fatty acid metabolic output level is the most noticeably increased. These findings are in keeping with the metabolic shift seen during cardiac development in vivo.

### 2.2. hiPSC-CMs Differentiation Leads to a Marked Increase in Fatty Acid Metabolism

To further understand the metabolism of differentiating hiPSC-CMs, we next investigated the changes in substrate metabolism rate during differentiation (Figure 2A,B). When the real-time rate of metabolism was tracked over a 6 h time period, the slowest rate of metabolism was seen in hiPSC fatty acid metabolism (3.2 AU). Following differentiation, there was a significant increase in the rate of metabolism for all substrate groups, with the largest fold change in metabolism rate seen for fatty acid substrates, increasing 18.0 fold from D0 to D30 (Figure 2C,D). Together, these analyses confirmed that a shift in metabolic reliance, representative of that seen in cardiac development and maturation, could be modelled with the differentiation of hiPSC-CMs. As with cardiac development, the greatest increase in metabolic output, and greatest change in metabolism rate, was seen in hiPSC-CM fatty acid metabolism. Additionally, this demonstrated the capability of hiPSC-CMs to be used in a real-time mitochondrial function assay, where shifts in specific substrate metabolism can be monitored.

### 2.3. ID Leads to a Metabolic Shift Only Partially Rescued by Transferrin

Next, we wanted to investigate if hiPSC-CMs could be used to detect differences in specific substrate metabolism during an induced diseased state such as ID. To this end, hiPSC-CMs were exposed to the iron chelator deferoxamine (DFO) for 4 days to simulate ID and we used transferrin for 2 days to rescue the iron deficiency phenotype given that transferrin supplementation is routinely used in iron replacement therapy. Characteristic alpha-actinin^+^ sarcomeres were observed in all conditions (Figure 3A), with morphological changes induced by the iron deficiency, including an apparent accumulation of intracellular granule build up (Appendix A). RT-qPCR identified alterations in genes involved in iron uptake within the iron-deficient hiPSC-CMs: including increased mRNA expression of transferrin receptor (TfRC), metal transporter-1 (SLC11A2), and Zip14 (SLC39A14), a mediator of non-transferrin-bound iron uptake (Figure 3B). Ferroportin mRNA (SlC40A1) was not increased in iron-deficient hiPSC-CMs (Figure 3B). Iron-deficient hiPSC-CMs had a significant reduction in contraction amplitude, producing contractions of 372.6 AU compared to the 2058.9 AU of control hiPSC-CMs (Figure 3C). Interestingly, analysis of calcium signalling genes, key for cardiomyocyte contraction, identified a significant increase in ryanodine receptor (RyR2), and decreased in calsequestrin 2 (CASQ2) mRNA expression within the ID cardiomyocytes (Figure 3B). This suggests that the contraction dysregulation during ID could be linked to poor calcium handling. Seahorse analysis further confirmed that ID induced a metabolism shift; iron-deficient hiPSC-CMs had a significantly reduced basal oxygen consumption rate (OCR) as well as maximal OCR after uncoupling with carbonyl cyanide-4-(trifluoromethoxy) phenylhydrazone (FCCP), indicating reduced mitochondrial respiratory function (Figure 3D). In contrast, the basal extracellular acidification rate (ECAR) was increased from 17.1 to 28.3 pmol/minute in iron-deficient hiPSC-CMs, indicating an increased reliance on glycolysis (Figure 3D). Of note, the transferrin rescue was unable to reverse the metabolic phenotype (Figure 3D). The decreased mitochondrial function observed in iron-deficient hiPSC-CMs did not correspond to a significant change in the relative protein expression of any of the mitochondrial ETC protein complexes (Figure 3E). However, it did lead to changes in gene expression, with several glycolysis genes being upregulated in iron-deficient hiPSC-CMs including pyruvate kinase (PKM), hexokinase II (HK2), and lactate dehydrogenase A (LDHA). The fatty acid metabolism genes acetyl-CoA carboxylase 1 (ACACA), and ATP citrate lyase (ACLY) did not increase in iron-deficient hiPSC-CMs (Figure 3B). Yet, the expression of the CD36 surface marker, a key fatty acid translocase shown to selectively identify matured and mitochondria-rich hiPSC-CMs [39], was down regulated in iron deficiency (Figure 3B). Interestingly, and potentially in response to the reduced mitochondrial function, the expression of mitochondrial biogenesis genes (PGC-1α and PPARα) was increased in iron-deficient hiPSC-CMs (Figure 3B).

Together, these data confirmed that an iron-deficient phenotype could be induced in hiPSC-CMs and demonstrated that ID leads to metabolic and functional changes in cardiomyocytes. Some of these phenotypes could be ameliorated by supplementation with transferrin but the rescue was incomplete as the metabolic shift was mostly unaltered in the rescue cardiomyocytes, highlighting the importance of understanding the metabolism of cardiomyocytes as well as what perturbs it and how it is perturbed.

### 2.4. ID Leads to a Unique Set of Metabolic Substrate Alterations, including Reduced TCA, Amino Acid and Fatty Acid Substrates

To further understand how ID affects cardiomyocyte metabolism we next performed a real-time mitochondrial function assay aiming to unveil the shifts in specific substrate utilisation upon ID induction (Figure 4A,B). The iron-deficient hiPSC-CMs had no change in peak metabolic output of any glycolytic substrates (Figure 4C). However, many non-glycolytic substrates were significantly reduced to ~50% peak metabolic output levels within iron-deficient hiPSC-CMs (Figure 4C). These substrates included components of the TCA cycle (d,l-isocitric acid, cis-aconitic acid, a-keto-glutaric acid, succinic acid, fumaric acid, a-keto-butyric acid), amino acid substrates (l-glutamic acid, l-glutamine, Ala-Gln, l-serine, l-ornithine, tryptamine), and fatty acid substrates (acetyl-l-carnitine, octanoyl-l-carnitine, palmitoyl-dl-carnitine chloride, d,l-b-hydroxy-butyric acid) (Figure 4C). When averaged, metabolic output of glycolytic substrates had the smallest reduction, only decreasing from 130.8 to 113.0 AU in ID (Figure 4D). Moreover, the average rate of metabolism was decreased for all substrate groups, but this reduction was also lowest in glycolytic substrates (0.84 fold), with fatty acid, TCA cycle and amino acid substrates, reducing further to 0.52, 0.44, and 0.39, respectively (Figure 4E). Collectively, these data confirmed that iron deficiency results in reduced mitochondrial respiration capacity, and an increased reliance on glycolytic metabolism within hPSC-CMs. Our results further uncovered which substrates have dysregulated metabolism upon iron deficiency. These included six TCA cycle substrates, five amino acid substrates, and four fatty acid substrates, which are good candidates for new therapeutic approaches.

### 2.5. Transferrin Rescue Restores TCA Cycle Substrate Metabolism, but Amino Acid and Fatty Acid Metabolism Remain Perturbed

Given that transferrin was unable to fully rescue the metabolic phenotype seen in ID hiPSC-CMs (Figure 3D), we next used the real-time mitochondrial function assay to determine which specific substrate metabolism was still affected within the transferrin hiPSC-CMs (Figure 5A). Of the substrates with metabolism significantly reduced in ID, those categorised as feeding directly into the TCA cycle had the greatest level of recovery, with an average group recovery of 59.9%. This resulted in the average metabolic output of TCA cycle substrates increasing from 108.08 AU in ID to 164.42 AU after rescue (Figure 5B). Within this group, the level of recovery varied, with the metabolic peak level of succinic acid (succinate) almost fully restored to 93%, whilst a-keto-butyric acid had the lowest level of recovery at 27% of control hiPSC-CMs (Figure 5A). Amino acid metabolism was rescued to a lesser extent, with an average group recovery of 29.8%. The amino acids l-glutamic acid, l-glutamine, Ala-Gln, l-serine, and l-ornithine, all displayed a moderate level of recovery (ranging between 20–45%), yet tryptamine metabolism showed no rescue (Figure 5A). Additionally, fatty acid metabolism had the lowest level of rescue, with no significant improvement in the level of carnitine metabolism. Of note, a trend was observed indicating higher levels of recovery may correspond to shorter fatty acid chain length (Figure 5A), although more investigation is required to confirm this.

A similar trend in rescue was observed when monitoring the changes in the rate of substrate metabolism (Figure 5C). When the real-time rate of metabolism was tracked over a 6 h time period, TCA cycle substrate metabolism had the greatest recovery in metabolic rate, increasing from 0.44 fold during iron deficiency to 0.87 fold of the control level following transferrin rescue (Figure 5C). This level of recovery was not seen in amino acid or fatty acid metabolism rate, with these remaining at 0.50 and 0.46 fold of the control levels, respectively (Figure 5C). Collectively, these data showed that the transferrin rescue leads to improvement on TCA cycle substrate metabolism, but that the amino acid and fatty acid metabolism were still perturbed to a greater extent, highlighting why this treatment does not usually allow for a complete recovery of the phenotype in patients.

## 3. Discussion

Here, we applied a novel mitochondrial function assay to assess the metabolism of a broad range of mitochondrial substrates in real time to study the effect of iron deficiency within hiPSC-CMs, and evaluate the extent of the metabolic rescue permitted by the supplementation of transferrin. This mitochondrial function assay revealed that ID leads to the dysregulation of six TCA cycle substrates, five amino acid substrates, and four fatty acid substrates, and that of these only some of the TCA substrate metabolites were rescued to levels close, but not equal, to those of control hiPSC-CMs. The low transferrin rescue seen at a metabolic level was not matched by an equal substandard rescue within other parameters measured such as contraction amplitude, where transferrin appeared to be more effective at restoring function. These data therefore highlighted that ID leads to a unique metabolite substrate fingerprint, revealing potential targets for more effective therapies than transferrin alone. A schematic summary of the observed alterations in substrate metabolism during iron deficiency and transferrin rescue is presented in Figure 6.

Metabolism of hiPSC-CMs has traditionally been studied using the Agilent Seahorse metabolic bioanalyser or mass spectrometry techniques, such as liquid chromatography–mass spectrometry (LC–MS) and capillary electrophoresis–mass spectrometry (CE–MS) [12,13,14,15,40]. While the first approach does not allow for specific substrate metabolism to be monitored, the second is time-consuming and uses static analysis of secreted metabolites. Mitochondrial function may also be assessed through use of a Clarke’s electrode or Oroborus, both of which allowing individual substrates or inhibitors to be tested for their effect on oxygen consumption. Similarly, radiolabelling of specific substrates can be used to interrogate specific metabolic pathways. Although these methods can provide valuable insights into substrate specific metabolism of cells, they are time-consuming and expensive and may therefore not be appropriate for the simultaneous study or screening of numerous modulators of metabolic pathways. Here, we tested a new approach, a real-time mitochondrial functional assay that monitored substrate utilisation within hiPSC-CMs [32]. This assay bypasses some of the shortcomings of both the Seahorse and mass spectrometry in that it enabled the study specific metabolite utilisation in real time. Importantly, we showed that this assay could be successfully used to study hiPSC-CMs metabolism during differentiation and in a disease context.

The mitochondrial function assay revealed that cells display a stage/cell type-specific profile in substrate utilisation during hiPSC-CM differentiation (Figure 1 and Figure 2), with early hiPSC-CMs (D6 and D10) displaying a lower maximal metabolic rate of all substrates compared to undifferentiated D0 hiPSCs, and later stage hiPSC-CMs (D20 and D30) displaying the greatest metabolic rates. This increase indicates that later stage hiPSC-CMs develop a more mature mitochondrial phenotype, recapitulating developmental events [16]. Indeed, as the cells differentiated, their metabolic output increased even when assayed in media devoid of any substrate. This may indicate that the differentiated cardiomyocytes have increased energy reserves, metabolising stored substrates such as glycogen and lipids. It remains to be seen how close this profile is to adult human cardiomyocytes, and it is likely that this increase does not produce fully mature cardiomyocytes, as previously reported for hiPSC-CMs [41]. Despite this potential immaturity, insights into the substrate utilisation during differentiation are evident.

Previous studies have revealed that cardiomyocytes have markedly higher expression of genes encoding enzymes involved in the TCA cycle compared to undifferentiated cells [42]. Here, we show TCA cycle substrates have the highest rate and peak metabolism in both hiPSCs and hiPSC-CMs. Rate and peak metabolism increased for TCA cycle substrates following cardiomyocyte differentiation. Of these, the biggest fold increase is observed for aconitic acid, a substrate catalysed by aconitases. This increased activity may in part be explained by the switch from aerobic glycolysis, supplying citrate for various anabolic processes including lipogenesis, towards oxidative metabolism through the TCA cycle. Mitochondrial aconitase acts as part of the citric acid cycle, whereas cytosolic aconitase is a trans-regulatory factor that controls iron homeostasis at a post-transcriptional level [43]. Both isoforms (mitochondrial and cytosolic) contain iron-sulphur clusters that allow binding to the hydroxyl groups of substrates [27]. It is therefore unsurprising that many studies have reported a significant loss of aconitase activity when iron deficiency is induced [43,44,45,46,47], and it is crucial that this activity is present in differentiated hiPSC-CMs to enable their use in modelling iron deficiency. Indeed, in this study, we observed a significant reduction in aconitic acid metabolic peak and rate during ID (Figure 4).

The metabolic shift from glycolysis towards oxidative phosphorylation is key to cardiomyocyte development and may be driving of the process, rather than simply a by-product [17]. Here, we showed the greatest increase in average metabolic output, and greatest change in metabolism rate, was in fatty acid metabolism following the differentiation of hiPSC-CMs (Figure 1 and Figure 2). The ability to monitor this shift over differentiation provides a valuable tool in the investigation of strategies that aim to promote the metabolic maturation of hiPSC-CMs [48]. Interestingly, and in contrast to these developmental events, ID has been shown to induce an opposing shift towards foetal-like glycolytic metabolism [29,30]. Our results support these findings and further demonstrate that this shift is not caused solely by an increase in glycolysis, but rather by a decrease in TCA, amino acid and fatty acid metabolism. This suggests that glycolysis may be the default metabolic mechanism of cardiomyocytes during iron deficiency and elucidates the metabolic perturbations subjacent to this deficiency.

Fatty acid uptake in myocardial tissue is tightly regulated and CD36 mediates approximately 70% of this essential process [49]. Changes in CD36 expression have been associated with various cardiovascular diseases, decreasing in pathological cardiac hypertrophy caused by ischaemia–reperfusion and pressure overload, and increasing in diabetic cardiomyopathy and atherosclerosis [50]. In the present study, we observed a significant reduction in CD36 expression in iron-deficient hiPSC-CMs, suggesting its likely involvement in the foetal-like glycolytic shift (Figure 3). The promoter region of CD36 contains PPARα response elements [51], and its expression levels during cardiac disease have been associated with levels of PPARα and PGC1α expression [50]. Interestingly, we observed expression levels of PPARα and PGC1α both increasing in iron-deficient hiPSC-CMs (Figure 3). The use of a PPARα agonist during differentiation has been shown to increase oxidative metabolism and expression of cardiac genes [52].

Previous work in animal models identified cardiac contraction pathways, particularly calcium signalling pathways, to be altered during ID [29]. In our hiPSC-CM model of ID, this reduced contractility appeared linked to the altered expression of the calcium signalling genes RYR2 and CASQ2 (Figure 3). A previous mouse model of iron deficiency reported a reduction in RYR2 expression in cardiac lysates [53]. In contrast, here we observed an increase in RYR2 in hiPSC-CMs, but a reduction in CASQ2 expression. A direct functional interaction between RyR2 and CASQ2 has been implicated in regulating Ca^2+^ release dynamics during cardiac contraction [54]. Ion concentration has been shown to influence this binding and it is therefore possible that the iron-deficient environment alters RyR2-CASQ2 interactions and regulatory expression.

A key finding of this study was the identification of substrates with reduced peak metabolic output levels in iron-deficient hiPSC-CMs, and the differing levels these recovered to during transferrin rescue (Figure 6). These substrates included components of the TCA cycle (d,l-isocitric acid, cis-aconitic acid, a-keto-glutaric acid, succinic acid, fumaric acid, and a-keto-butyric acid), amino acid substrates (l-glutamic acid, l-glutamine, Ala-Gln, l-serine, l-ornithine, and tryptamine), and fatty acid substrates (acetyl-l-carnitine, octanoyl-l-carnitine, palmitoyl-dl-carnitine chloride, and d,l-b-hydroxy-butyric acid). The high recovery levels in the metabolism of many TCA cycle substrates may contribute to the functional rescue of hiPSC-CM contractility and mitochondrial respiration (Figure 3). Thus, clinically, ID may perpetuate the metabolic inflexibility and shift towards foetal metabolism observed in heart failure, and may, at least in part, explain adverse outcomes in cardiovascular disease patients which also suffer from ID [25,26,27,28,55].

The identification of substrates with low recovery levels provide targets for new biomarker discovery and additional pharmaceutical strategies. Notably, the amino acid tryptamine showed no level of recovery following transferrin rescue (Figure 5). Tryptamine is produced through tryptophan metabolism, levels of which, have previously been shown to correlate with biomarkers of iron metabolism in individuals with and without iron deficiency [56]. Of the fatty acids, none of the short-(acetyl-l-), medium-(octanoyl-l-), or long-(palmitoyl-dl-) chain carnitine forms showed a significant level of metabolic recovery following transferrin supplementation (Figure 5). Identification of potential metabolomic biomarkers could offer a more accurate phenotype and prognostic value compared to measurement of conventional clinical biomarkers alone, thereby advancing the development of precision medicine and the identification of new pharmaceutical targets [57].

Collectively, this study showed that the metabolic shift from glycolysis towards oxidative phosphorylation seen during hiPSC-CM differentiation was lost during ID and can, at least partly, be restored through iron supplementation with transferrin. The application of hiPSC-CMs within a mitochondrial function assay, capable of measuring the real-time metabolism of specific substrates, allowed us to uncover the unique metabolic fingerprint of ID cardiomyocytes and identify the metabolic shortcomings of the current transferrin treatment. More broadly, this study highlights the potential of this novel metabolic function analysis technique for the screening of novel therapeutics aimed at modifying metabolic activity and functional capacity of cardiomyocytes.

## 4. Methods

### 4.1. hiPSC Culture and Cardiomyocyte Differentiation

The WTC-11 male hiPSC line (GM25256), and the derived fluorescent reporter lines (AICS-0011 and AICS-0075-085), were obtained from Allen Cell Collection (available through Coriell Institute), and were maintained and grown in Essential 8 (E8) media (Gibco, Life Technologies, Leicestershire, UK) on Vitronectin (Gibco, Life Technologies) coated surfaces in a humidified atmosphere (5% CO_2_) at 37 °C. The media was replaced every 24 h. When hiPSCs reached 90% confluency they were dissociated using TrypLE Express (Gibco, Life Technologies) following the manufacturer’s protocol. Cells were seeded at 20,000 cells/cm^2^ with E8 supplemented with a Rho kinase inhibitor (Y-27632, Abcam, Cambridge, UK).

The cardiomyocyte differentiation protocol was adapted from Burridge et al. 2014 [58] in order to promote left ventricle cardiomyocyte differentiation (Dark et al. manuscript in preparation). A patent has been submitted for this method: WO2020245612.

To induce iron deficiency in hiPSC-CMs, cells were treated with 30 µM deferoxamine (Sigma-Aldrich, Gillingham, UK) in CDM3 medium for 4 days as previously described [30]. CDM3 medium consisted of RPMI (Gibco, Life Technologies), 500 μg/mL rice-derived recombinant human albumin (Sigma-Aldrich, Gillingham, UK) and 213 μg/mL L-ascorbic acid 2-phosphate (Tocris). To restore intracellular iron levels, cells were incubated with CDM3 medium supplemented with 5 µg/mL partially saturated transferrin (Sigma-Aldrich, Gillingham, UK) for 2 days [30]. During experiments, medium was replenished daily for all conditions.

### 4.2. Immunocytochemistry and Imaging

For immunostaining, hiPSC-CMs were plated on thin black bottom 96 well plates (Corning). Cells were washed using PBS followed by fixing using 5% PFA (Sigma-Aldrich) for 15 min followed by 2x PBS washes. Fixed cells were then washed for 15 min in 0.1% Triton X-100 (Sigma-Aldrich, Gillingham, UK) in PBS followed by 3x further washes in PBS. Fixed cells were then blocked in 4% Goat Serum (Sigma-Aldrich, Gillingham, UK) in PBS for 1 h followed by a further 5 min PBS wash. Primary antibody was added to the plates, diluted in 4% Goat Serum in PBS (Alpha Actinin 1:800, Abcam) and left overnight at 4 °C on a rocking plate. The following day fixed cells were washed with 0.1% Tween (Sigma-Aldrich, Gillingham, UK) in PBS (3 × 10 min) followed by secondary antibody staining [Anti-Mouse Alexa Fluor^®^ 488 (Abcam, Cambridge, UK) diluted 1:500 in 4% Goat Serum (Sigma-Aldrich, Gillingham, UK) in PBS] for 1 h at room temperature. The fixed cells were then washed 3 times with 0.1% Tween-PBS (Sigma-Aldrich, Gillingham, UK) followed by incubation in PBS supplemented with DAPI (Tocris, Abingdon, UK) at 1:500 for 10 min. A further wash was performed using PBS after which the fixed cells were submerged in PBS and the plate wrapped in foil prior to imaging using the EVOS M5000 imaging system.

### 4.3. Gene and Protein Expression Analysis

RNA was extracted from cells using the NucleoSpin RNA Kit (Macherey-Nagel) following the manufacturers’ provided protocols. Reverse transcription was performed using the Ultrascript 2.0 cDNA Synthesis Kit (PCR Biosystems, London, UK) following the manufacturers’ protocol. Real-time quantitative PCR was performed using 1 μL of the cDNA reaction with 2.5 μL SYBR Green PCR Master Mix (PCR Biosystems, London, UK), 0.5 μL primer master mix and 1 μL H_2_O. Reactions were performed on Roche LightCycler 480 Multiwell Plates in a Roche LightCycler 480 II. All qPCR reactions were performed in triplicate, normalised to housekeeper gene GAPDH and assessed using a comparative C_t_ method. Primer sequences are shown in Table 1.

Protein was extracted from cells using 100 μL of SDS-PAGE sample buffer supplemented with proteinase inhibitor (at 10 μL/mL). Cell lysate were run on a 4–15% gradient polyacrylamide gel (Bio-Rad), transferred to a PVDF membrane (Bio-Rad) using the Trans-Blot Turbo transfer system (Bio-Rad) following the manufacturers’ protocol. For immunofluorescent westerns blots membranes were blocked in Intercept^®^ blocking buffer (LI-COR) for 1 h. Blocking buffer was replaced with fresh Intercept^®^ blocking buffer supplemented with 0.2% Tween (Sigma-Aldrich, Gillingham, UK) and primary antibodies at 1:200 (Total OXPHOS Human Cocktail, ab10411, Abcam, Cambridge, UK) and incubated overnight at 4 °C. Membranes were then washed with 0.1% Tween in TBS (4 × 7.5 min), followed by incubation with secondary antibody (anti-Mouse, LI-COR) at 1:10,000 in blocking buffer supplemented with 0.2% Tween and 0.02% SDS for 1 h at room temperature. After washing with 0.1% Tween in TBS (4 × 7.5 min), membranes were imaged using the LI-COR Odyssey^®^ imaging system. For chemiluminescent Western blots membranes were blocked in 5% Milk in 0.1% Tween in PBS (PBST) for 1 h, blocking solution was replaced with fresh 5% Milk in PBST with primary antibodies at 1:200 (Total OXPHOS Human Cocktail, ab10411, Abcam) incubated overnight at 4 °C. Membranes were then washed with PBST (4 × 7.5 min) followed by incubation with secondary antibody (anti-Mouse, Invitrogen) at 1:5000 in 5% Milk in PBST for 1 h. After washing with PBST (4 × 7.5 min), membranes were imaged using LumiBlue Extra substrate (Expedeon) on a CCD camera.

### 4.4. Contraction Analysis

Videos of hiPSC-CM contraction were recorded using a ClonerAlliance Box Turbo (ClonerAlliance) attached to the ThermoFisher EVOS M5000 imaging system, with cells maintained in a humidified atmosphere (5% CO_2_) at 37 °C in an onstage incubator during recording. Videos were recorded for 20 s, then cropped and converted to TIFF stacks using Adobe Photoshop CC2020. Compiled TIFF stacks were processed using the MUSCLEMOTION plugin with ImageJ V1.53 [59]. Output data values were stored in a Microsoft Excel sheet, any false peak contractions were removed [59], and all contraction amplitude results were averaged within the appropriate group and analysed using Prism Graph Pad V5.

### 4.5. Metabolic Analysis

For Seahorse metabolic flux assay, hiPSC-CMs were seeded onto Vitronectin coated Seahorse XFp cell culture miniplates (Agilent, Santa Clara, CA, USA) to create a confluent monolayer 4 days prior to starting the iron deficiency assay. Cells were maintained in a humidified non-CO_2_ incubator until the start of the assay. XFP flux cartridges (Agilent) were pre-hydrated in XF Calibrant (Agilent, Santa Clara, CA, USA) overnight at 37 °C prior to running seahorse experiment. Flux cartridges were loaded with Oligomycin (2 µM), carbonyl cyanide-4-(trifluoromethoxy) phenylhydrazone (FCCP) (1 µM), and Rotenone (0.5 µM) according to manufacturer’s instructions. The assay was performed in Seahorse XF Media, supplemented with 1 mM Sodium pyruvate, 2 mM l-Glutamine, and 10 mM Glucose. Oxygen consumption rate and basal extracellular acidification rate values were obtained using the XFp Mito Stress Kit (Agilent, Santa Clara, CA, USA). Following the seahorse assay XFp cell culture miniplates were fixed for 10 min and stained with DAPI for 15 min. Each well cell count was calculated using automated nuclei counting as part of the in-built software on the EVOS M5000 imaging system. All results were normalised to output cell count.

For mitochondrial substrate functional assay, MitoPlates (Biolog) were prepared following manufactures instructions, 30 µL assay mix containing 50 µg/mL Saponin (Sigma-Aldrich) were pipetted into the wells and incubated at 37 °C for 1 h. Cells were dissociated, resuspended in 1x MAS Buffer (Biolog) creating a final concentration of 2 × 10^6^ cell/mL following which 30 µL of cell suspension were pipetted into each well of the MitoPlate. The MitoPlate was read for 6 h with 5 min intervals at OD590 using a OmniLog plate reader. OmniLog peak and rate values were calculated on Data Analysis 1.7 software before being transferred to a Microsoft Excel sheet for reformatting.

### 4.6. Data Interpretation

After the 6 h time period the final OD590 reading was subtracted against the initial reading to calculate the change in dye reduction from electrons flowing into and through the ETC, this value is presented as peak metabolic output. The OmniLog data analysis 1.7 software calculates the maximal rate change from the gradient of the OD590 output during the 6 h time period. These data are either presented as raw OD590 (AU) or as a relative fold change to control conditions.

### 4.7. Quantification and Statistical Analysis

Statistical analysis was carried out using Prism Graph Pad. Differences among groups were considered significant when the probability value, *p*, was less than 0.05 (*), 0.01 (**), or 0.001 (***). No statistical methods were used to predetermine sample size. For statistical comparison of more than two groups the two-way ANOVA was performed.

## Figures and Tables

**Figure 1 metabolites-12-00009-f001:**
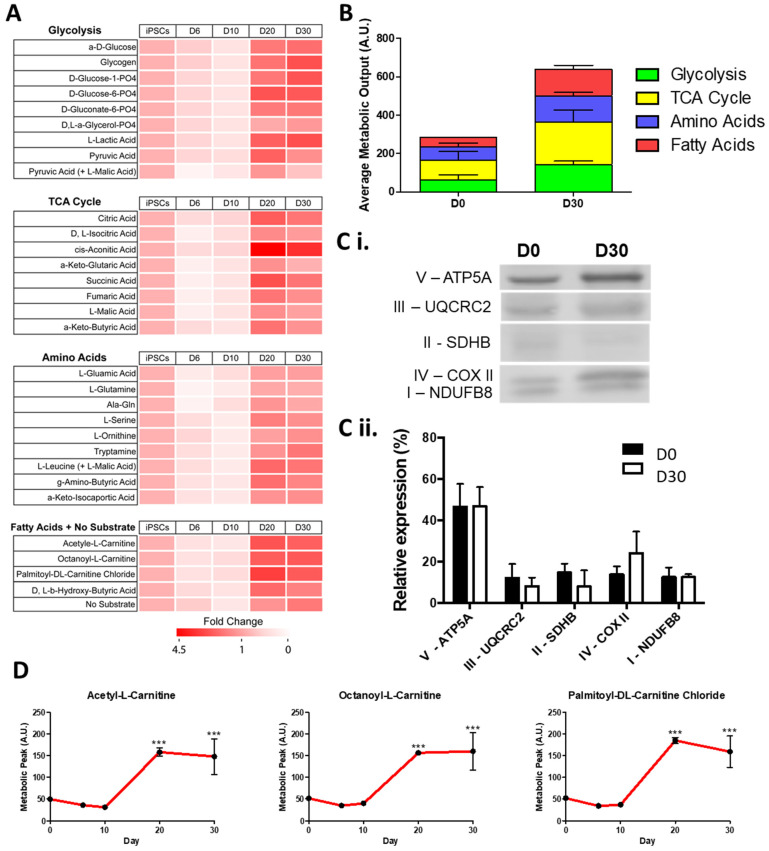
Change in maximum metabolic capacity during the differentiation of hiPSC-CMs. In (**A**), relative fold change in maximal metabolic capacity of glycolysis, TCA cycle, amino acid, and fatty acid substrates at day 6, 10, 20, and 30 of hiPSC-CMs differentiation normalised to undifferentiated day 0 hiPSCs is shown as a heat map. In (**B**), the maximum peak is averaged for each of the glycolysis, TCA cycle, amino acid, and fatty acid substrate groups. In (**C**), a representative Western blot of subunits of mitochondrial complexes I–V at day 0 and day 30 of differentiation is shown (**Ci**) normalised as a relative level of total expression (**Cii**). In (**D**), the metabolic peak levels of short-(acetyl-l-), medium-(octanoyl-l-), and long-(palmitoyl-dl-) chain carnitine forms is shown over the differentiation timepoints. n = 3. Significance was determined by two-way ANOVA, *** *p* < 0.001.

**Figure 2 metabolites-12-00009-f002:**
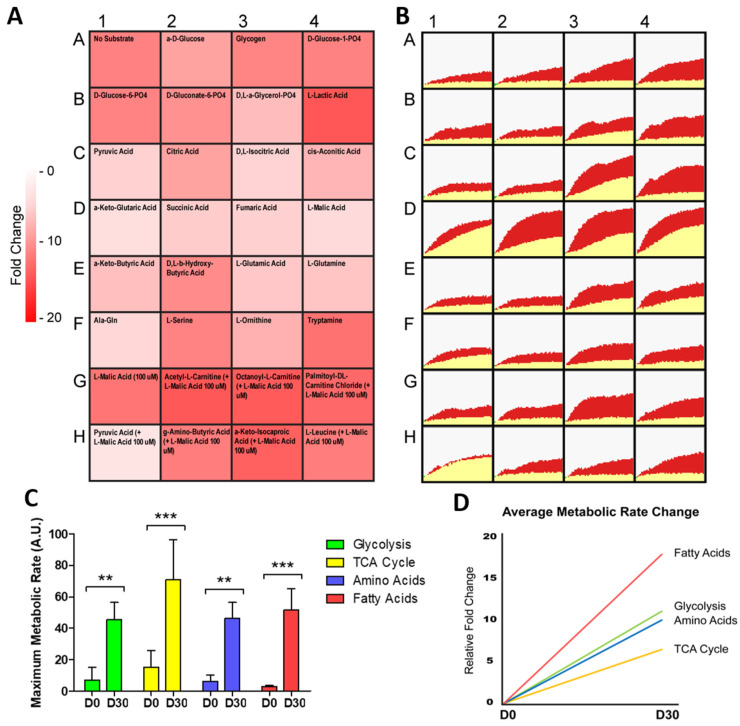
Change in metabolic rate during the differentiation of hiPSC-CMs. In (**A**), relative fold change in the rate of substrate metabolism for day 30 hiPSC-CMs normalised to undifferentiated day 0 hiPSCs is shown as a heat map. In (**B**), corresponding representative rate traces are shown for both day 30 hiPSC-CMs (shown in red) and day 0 hiPSCs (shown in yellow), highlighting the increase in rate. In (**C**), the maximum metabolic rate for substrates categorised in each of the glycolysis, TCA cycle, amino acid, and fatty acid substrate groups was averaged and plotted for day 30 hiPSC-CMs and day 0 hiPSCs. n = 3. In (**D**), the relative fold change in the grouped substrate average is plotted for day 30 hiPSC-CMs. Significance was determined by two-way ANOVA, ** *p* < 0.01, *** *p* < 0.001.

**Figure 3 metabolites-12-00009-f003:**
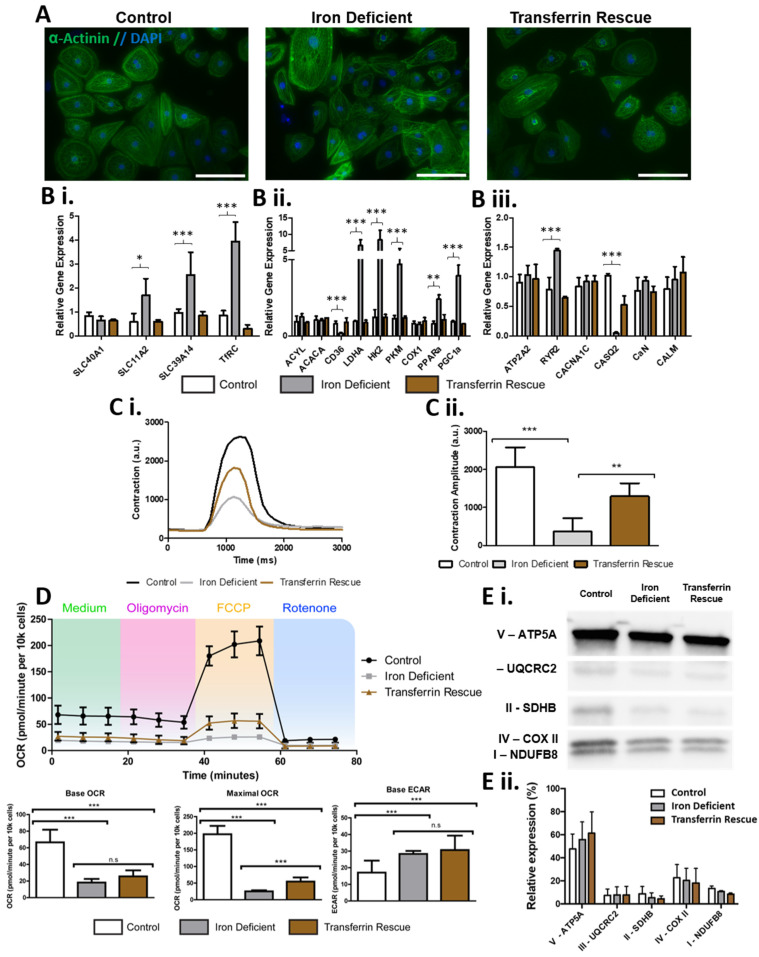
Effects of iron deficiency and transferrin rescue on hiPSC-CMs. In (**A**), alpha-actinin staining (green) and DAPI (blue) is shown for hiPSCs-CMs with iron depletion and transferrin rescue showing high purities of cardiomyocytes. In (**B**), qPCR analysis of iron-deficient and transferrin-rescued hiPSC-CMs for genes involved in (**Bi**) iron uptake, (**Bii**) glycolysis and fatty acid metabolism, and (**Biii**) Ca2+ signalling, with expression levels normalised to control hiPSC-CMs. In (**C**), example traces of hiPSC-CM contractions are shown (**Ci**), with calculated amplitude quantification (**Cii**) n = 6. In (**D**), Cardiac bioenergetics analysis using the Seahorse platform is shown with quantification of base and maximal oxygen consumption rate, base extracellular acidification rate. In (**E**), a representative Western blot of subunits of mitochondrial complexes I–V in control, iron-deficient, and transferrin rescued hiPSC-CMs is shown (**Ei**) normalised as a relative level of total expression (**Eii**). n = 3. Scale bar is 125 µm. Significance was determined by two-way ANOVA, * *p* < 0.05, ** *p* < 0.01, *** *p* < 0.001.

**Figure 4 metabolites-12-00009-f004:**
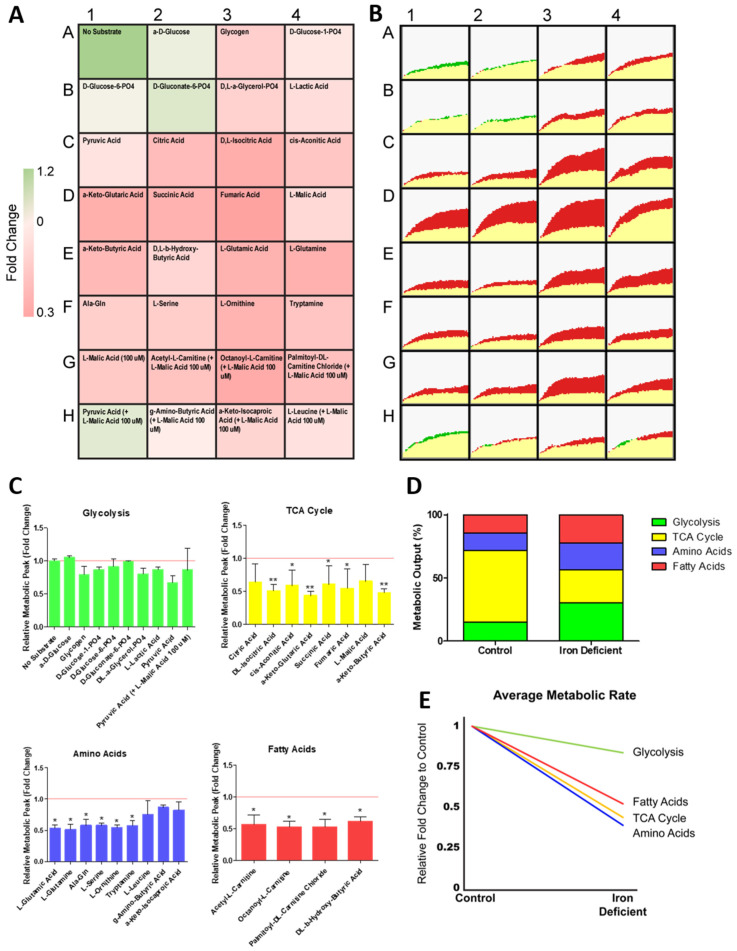
Change in metabolism during hiPSC-CM iron deficiency. In (**A**), relative fold change in the rate of substrate metabolism for iron-deficient hiPSC-CMs normalised to control hiPSC-CMs is shown as a heat map. In (**B**), corresponding representative rate traces are shown for both iron-deficient hiPSC-CMs (shown in yellow/green) and control hiPSC-CMs (shown in yellow/red). (**C**) substrate maximum metabolic capacity for iron-deficient hiPSC-CMs normalised to control hiPSC-CMs. n = 3. In (**D**), the maximum metabolic peak is averaged for each of the glycolysis, TCA cycle, amino acid, and fatty acid substrate groups. In (**E**), the relative fold change in the rate of grouped substrate average is plotted for iron-deficient hiPSC-CMs compared to control. Significance was determined by two-way ANOVA, * *p* < 0.05, ** *p* < 0.01.

**Figure 5 metabolites-12-00009-f005:**
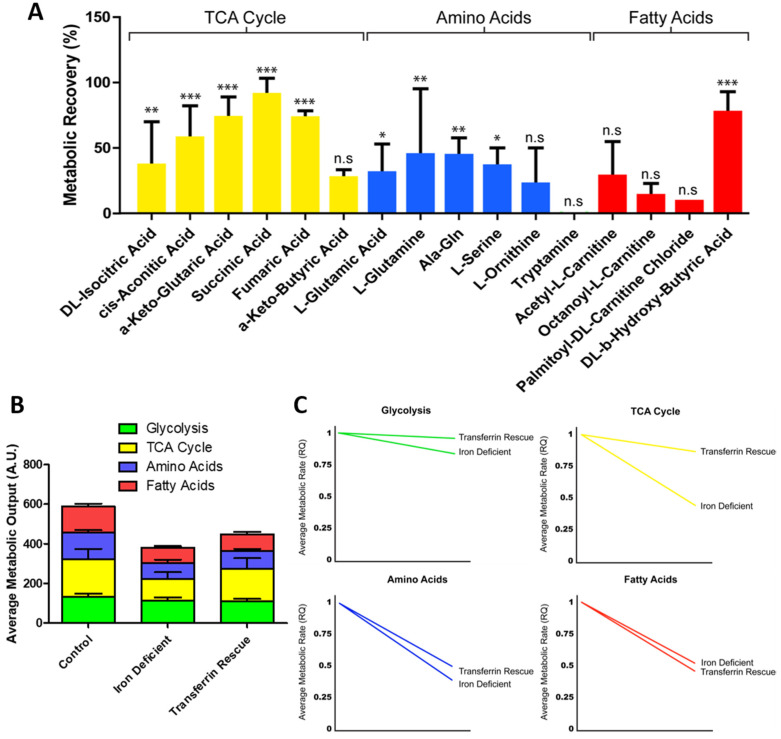
Change in metabolism during transferrin rescue of iron-deficient hiPSC-CMs. Substrate maximum metabolic capacity for transferrin-rescued hiPSC-CMs normalised to iron-deficient hiPSC-CMs. n = 3 (**A**). In (**B**), the maximum metabolic peak is averaged for each of the glycolysis, TCA cycle, amino acid, and fatty acid substrate groups and plotted as a percentage of total metabolic output. In (**C)**, the relative fold change in the rate of grouped substrates is plotted for iron-deficient and transferrin-rescued hiPSC-CMs compared to control. Significance was determined by two-way ANOVA, * *p* < 0.05, ** *p* < 0.01, *** *p* < 0.001.

**Figure 6 metabolites-12-00009-f006:**
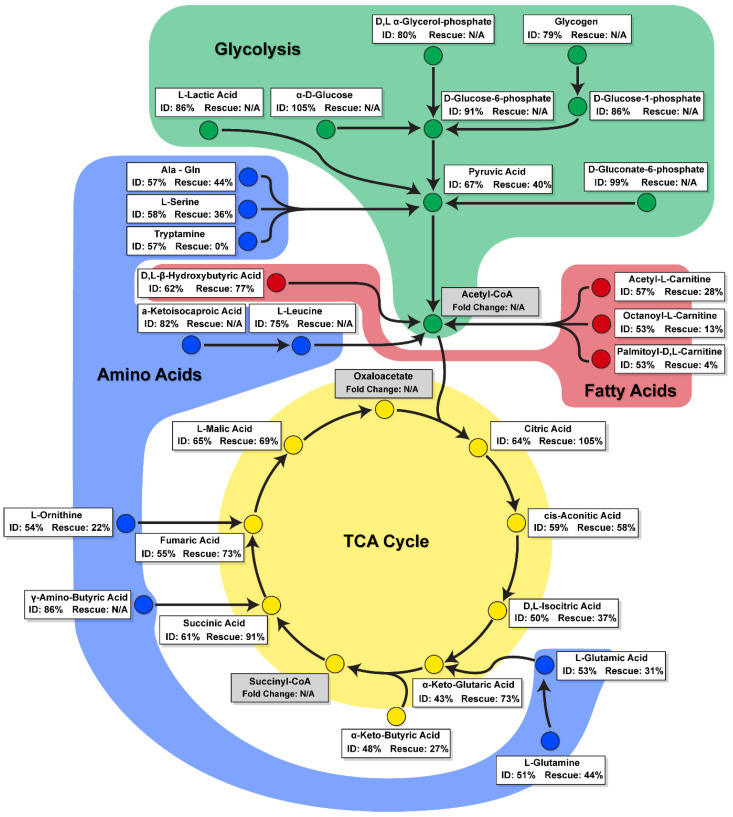
A schematic summary of change in substrate metabolism during iron deficiency and transferrin rescue. Substrates are categorised as involved in glycolysis (green), fatty acid (red), amino acid (blue), or TCA cycle (yellow) metabolism pathways. Percentage change for ID is shown compared to control. Percentage change for Rescue is shown compared to ID, with substrates that did not significantly change during ID stated as N/A.

**Table 1 metabolites-12-00009-t001:** Primer sequences used for SYBR Green RT-qPCR.

Gene Primer	Forward Sequence (5′-3′)	Reverse Sequence (5′-3′)
GAPDH	GTCTCCTCTGACTTCAACAGCG	ACCACCCTGTTGCTGTAGCCAA
SLC11A2	GGACTGTGGGCATACGGTAA	ACACTGGCTCTGATGGCTAC
SLC39A14	TTGCGCTAGCTGGAGGAATG	TGGAATCAAGATGCTGCCCTT
SLC40A1	CTAGTGTCATGACCAGGGCG	CACATCCGATCTCCCCAAGT
TFRC	TGGCAGTTCAGAATGATGGA	AGGCTGAACCGGGTATATGA
ACACA	TTCACTCCACCTTGTCAGCGGA	GTCAGAGAAGCAGCCCATCACT
ACLY	GCTCTGCCTATGACAGCACCAT	GTCCGATGATGGTCACTCCCTT
PKM	ATGGCTGACACATTCCTGGAGC	CCTTCAACGTCTCCACTGATCG
HK2	GAGTTTGACCTGGATGTGGTTGC	CCTCCATGTAGCAGGCATTGCT
LDHA	GGATCTCCAACATGGCAGCCTT	AGACGGCTTTCTCCCTCTTGCT
PGC-1a	AGGCTAGTCCTTCCTCCATGC	GTTGGCTGGTGCCAGTAAGAG
COX1	TCCTTATTCGAGCCGAGCTG	GGGCTGTGACGATAACGTTG
RYR2	CCTTGCCTGAGTGCAGTTG	TTGAGGTATCAACAGGTTGTGG
CASQ2	AGCTTGTGGAGTTTGTGAAG	GGATTGTCAGTGTTGTCCC
CALM1	TGCGGAAGTTAGGAGTGCTG	GCACAGCATAATGGAAGGCG
CaN	AGTAACTTTCGAGCCAGCCC	GGGGGTCTGACCACAAGATG
ATP2A2	CTCGGATCCAACACTACAGGTGTTGAATGG	CGGAATTCATGCGCAGTGATAAATTGAC
CACNA1C	AAGGCTACCTGGATTGGATCAC	GCCACGTTTTCGGTGTTGAC

## Data Availability

The data presented in this study are available in the article and Appendix A.

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
