# Peer review of "Modelling Metabolic Shifts during Cardiomyocyte Differentiation, Iron Deficiency and Transferrin Rescue Using Human Pluripotent Stem Cells"

_metabolites, 2021, doi:10.3390/metabo12010009_

Round 1

Reviewer 1 Report

Interesting study focusing on cardiomyocyte metabolism and ID induced metabolic changes. Major limitation is the use of one cell model and not providing deeper molecular insights under ID and transferrin treatments.

Major comments:

- Can the authors perform some of these experiments in neonatal cardiomyocytes isolated from postnatal days (marking differentiation window) in rodents? That would help expand the conclusions driven from these experiments done using ID and transferrin for example. Currently, these were done on iPS-CMs and therefore data on primary cells would strengthen their conclusions.

- Data are interesting. Therefore, more insights at the level of gene expression changes in iPS-CMs under ID and transferrin treatment would be a great addition to the community. This will provide an overall view on pathways regulated in these cells under ID and transferrin rescue conditions. Because the treatment of transferrin only rescued ID deficiency partially, performing an RNA-seq would provide deeper insights on understanding the basis of ID.

- Fatty acids are known to induce hypertrophy in iPS-CMs. Do the authors observe any such sort of changes between the timepoints they studied? What are the effects of ID and transferrin treatment on cell morphology (size etc).

Author Response

Response to Reviewer 1

We would like to thank the reviewer for taking the time to read and review our manuscript and we are pleased that they found it an interesting study. We have revised our manuscript using the track changes function in Word, and have highlighted changes in the text in yellow to make them easy to find. The line numbers referred to are accurate when using the ‘simple markup’ view of the track changes. Our responses to the specific raised points are listed below in red text.

Interesting study focusing on cardiomyocyte metabolism and ID induced metabolic changes. Major limitation is the use of one cell model and not providing deeper molecular insights under ID and transferrin treatments.

Major comments:

  • Can the authors perform some of these experiments in neonatal cardiomyocytes isolated from postnatal days (marking differentiation window) in rodents? That would help expand the conclusions driven from these experiments done using ID and transferrin for example. Currently, these were done on iPS-CMs and therefore data on primary cells would strengthen their conclusions.

Despite a high degree of genomic and physiologic similarities between rats and humans, species specific functional differences within metabolic pathways have been described in detail (Blais et al. 2017). Indeed, studies have shown that different rodent model systems vary in their iron metabolism (Shannahan et al. 2010). As such, although we believe it would be interesting to investigate changes in rodent model cells system, hiPSC-CMs are the most appropriate system for this special edition.  

Blais, E., Rawls, K., Dougherty, B. et al. Reconciled rat and human metabolic networks for comparative toxicogenomics and biomarker predictions. Nat Commun 8, 14250 (2017).

Shannahan, J. H. et al. Pulmonary oxidative stress, inflammation, and dysregulated iron homeostasis in rat models of cardiovascular disease. J. Toxicol. Environ. Health A 73, 641–656 (2010).

  • Data are interesting. Therefore, more insights at the level of gene expression changes in iPS-CMs under ID and transferrin treatment would be a great addition to the community. This will provide an overall view on pathways regulated in these cells under ID and transferrin rescue conditions. Because the treatment of transferrin only rescued ID deficiency partially, performing an RNA-seq would provide deeper insights on understanding the basis of ID.

We are pleased the reviewer finds our data interesting. The targeted qRT-PCR in Figure 3B presents gene expression changes in (i) iron uptake, (ii) glycolysis and fatty acid metabolism, and (iii) Ca2+ signaling pathways. We fully agree that RNA-seq would further expand the insights into these pathway and uncover others. As such, we have future plans to carry out this work when the necessary funding has been acquired, but we believe that such investigations are beyond the scope of the current manuscript. 

  • Fatty acids are known to induce hypertrophy in iPS-CMs. Do the authors observe any such sort of changes between the timepoints they studied? What are the effects of ID and transferrin treatment on cell morphology (size etc).

This is an interesting point and is also an avenue we initially explored. The composition of culture media, and therefore the pathways induced by the metabolism of these, has been shown to influence cell morphology. Supplementation of culture media with fatty acids has been shown to induce hypertrophy (Yang et al. 2019), whilst high-glucose media has also been shown to significantly increase cell size (Kwong-Man et al. 2018). As such, modification of metabolism further towards either of these pathways may influence cell size. In our 2D culture system, we did not observe a change in cell size induced by iron deficiency. However, we did observe morphological changes within the DFO treated cells, with evidence of intracellular granule build up. We have now included brightfield images of the treated cells in a supplementary figure (Supplementary Figure 1) to demonstrate this and refer to this figure on line 165.

‘with morphological changes induced by the iron deficiency, including an apparent accumulation of intracellular granule build up (Supplementary Figure 1).’       

Yang et al. Fatty Acids Enhance the Maturation of Cardiomyocytes Derived from Human Pluripotent Stem Cells. Stem Cell Reports. Volume 13, Issue 4, Pages 657-668, (2019).

Kwong-Man et al. Empagliflozin Ammeliorates High Glucose Induced-Cardiac Dysfuntion in Human iPSC-Derived Cardiomyocytes. Sci Rep.; 8: 14872. (2018).

Reviewer 2 Report

In this paper, the authors use a mitochondrial function assay to profile mitochondrial changes as iPSCs differentiate towards cardiomyocytes and then to investigate changes in the iPSC-CM when cultured in iron-deficient media. The upregulation of oxidative metabolism during differentiation is well established but the effect of iron depletion is novel as is the use of the Biolog assay. Although it provides a large amount of interesting information, the Biolog assay needs to be interpreted with care. As I understand it, it is more a measure of enzyme activity than of substrate utilisation, in that in each well only a single substrate is presented so the ability of the cell to metabolise that substrate through some part of the TCA cycle/ETC is quantified. There is no 'choice' of substrate and many of the substrates tested bypass the process of uptake into the cells. This is very evident in the comparison between the Seahorse data and mitochondrial data in figs 3 and 4. The Seahorse data clearly shows an increase in glycolysis in the iron-deficient cells, yet the mitochondrial data showed no difference between control and iron-deficient cells treated with glycolytic substrates. This is because the iron-deficient cells retain the same ability to perform glycolysis despite the lack of iron (possibly because of the absence of enzymes with iron sulphur clusters in the glycolysis pathway) but they have to upregulate glycolysis due to impaired efficiency of the oxidative pathways. The authors tend to discuss their results as if this assay were measuring the whole metabolic pathway for glucose, fatty acids etc and need to rephrase areas of their interpretation of the data to ensure they are not making invalid assumptions. Furthermore, there may be more that they could draw from the data if they compare flux before and after various points in the TCA cycle or electron transport chain for example.

A brief introduction to the technique would be useful so that the reader understands what is being measured in this relatively new technique. For example - how is the percentage contribution to metabolic output quantified? This isn’t straightforward with so many pathways being assessed. Data interpretation could be presented in a data supplement perhaps. The manufacturer’s website doesn’t make this clear so more details are needed.

A bit more background to the role of iron in metabolism would be helpful, such as the significance of iron-sulphur clusters in the electron transport chain.

In Figure 1 it is clear from the graphs in 1D that the drop in max metabolic rate is small compared to the increase at day 20 – this isn't clear from the text.

In Fig 1Cii relative expression of various mitochondrial proteins are presented. Does the total expression change - I would have expected mitochondrial number to increase and so total expression to increase?

Line 117: You say ‘In both D0 hiPSCs 117 and D30 hiPSC-CMs, the highest rates of metabolism were observed with direct TCA cycle substrates, with rates of 16.9 A.U. and 75.0 A.U., respectively’ Is this statistically significant, no stats are given between groups? Also, what does this tell us, since the TCA cycle is used in metabolism of glucose, amino acids and fatty acids? 

Figure 2: the change seen in ‘no substrate’ is interesting. Presumably the cells are in substrate-free media and so are metabolising stored substrates such as glycogen and lipids? They must be using something since there is a change as the cells differentiate.

Line 140 add in number of days of transferrin rescue to main text as well as having it in the methods.

Line 154 What substrates were available in the Seahorse experiment – were there fatty acids in the media or just glucose?

Line 208 You say ‘Our results further uncovered which substrates are dysregulated upon iron deficiency.’ This should refer to pathways rather than substrates or say 'Our results further uncovered which substrates have dysregulated metabolism upon iron deficiency.'

Line 276 You should mention other oxygen consumption assays, such as the Clarke electrode or Oroborus, which do allow substrate specific oxygen consumption measurements to be made, and radiolabelled experiments which allow substrate utilisation to be measured. All these give different information that can be combined to give a more complete picture, to which the mitochondrial measurements made here would provide valuable additional insights.

In Line 302 you don’t discuss why aconitase activity might increase with differentiation. This could be due to the switch from aerobic glycolysis in iPSCs and early differentiation, where glycolytic substrates exit the TCA cycle at citrate for de novo lipogenesis, to oxidative metabolism with full use of the TCA cycle.

Thankyou for figure 6 – it is very clear and helpful.

Author Response

Response to Reviewer 2

We would like to thank the reviewer for taking the time to read and review our manuscript and we are pleased that they found it an interesting study. We have revised our manuscript using the track changes function in Word, and have highlighted changes in the text in yellow to make them easy to find. The line numbers referred to are accurate when using the ‘simple markup’ view of the track changes. Our responses to the specific raised points are listed below in red text.

In this paper, the authors use a mitochondrial function assay to profile mitochondrial changes as iPSCs differentiate towards cardiomyocytes and then to investigate changes in the iPSC-CM when cultured in iron-deficient media. The upregulation of oxidative metabolism during differentiation is well established but the effect of iron depletion is novel as is the use of the Biolog assay. Although it provides a large amount of interesting information, the Biolog assay needs to be interpreted with care. As I understand it, it is more a measure of enzyme activity than of substrate utilisation, in that in each well only a single substrate is presented so the ability of the cell to metabolise that substrate through some part of the TCA cycle/ETC is quantified. There is no 'choice' of substrate and many of the substrates tested bypass the process of uptake into the cells. This is very evident in the comparison between the Seahorse data and mitochondrial data in figs 3 and 4. The Seahorse data clearly shows an increase in glycolysis in the iron-deficient cells, yet the mitochondrial data showed no difference between control and iron-deficient cells treated with glycolytic substrates. This is because the iron-deficient cells retain the same ability to perform glycolysis despite the lack of iron (possibly because of the absence of enzymes with iron sulphur clusters in the glycolysis pathway) but they have to upregulate glycolysis due to impaired efficiency of the oxidative pathways. The authors tend to discuss their results as if this assay were measuring the whole metabolic pathway for glucose, fatty acids etc and need to rephrase areas of their interpretation of the data to ensure they are not making invalid assumptions. Furthermore, there may be more that they could draw from the data if they compare flux before and after various points in the TCA cycle or electron transport chain for example.

We would like to thank the reviewer for their fair and useful comments. We agree that our use of a novel assay requires more detailed introduction and explanation. We also agree that some of our results/ text required rephrasing as the Biolog assay did not provide a ‘choice’ of substrate and therefore some of the wording could be more appropriate. We have changed this throughout. The lack of substrate ‘choice’ was something we had already discussed ourselves, and as such we had already commenced projects that are trying to further adapt the assay to overcome this limitation.

  • A brief introduction to the technique would be useful so that the reader understands what is being measured in this relatively new technique. For example - how is the percentage contribution to metabolic output quantified? This isn’t straightforward with so many pathways being assessed. Data interpretation could be presented in a data supplement perhaps. The manufacturer’s website doesn’t make this clear so more details are needed.

We have added a paragraph to the introduction explaining the technique and referencing several recent papers that have used this technique. This paragraph begins on line 92.

‘In this study we use a novel, dynamic 96-well microplate assay that is based on colorimetric changes representing electron fluxes through the mitochondrial ETC. This assay assesses a panel of 31 different substrates that are provided to cells that have been permeabilized to ensure uptake. These substrates, utilized in various metabolic pathways and facilitated by specific enzymes and co-factors, provide either nicotinamide adenine dinucleotide (NADH) or flavin adenine dinucleotide (FADH2), ultimately supplying electrons to complex I or complex II, respectively. Electrons then transition through the ETC and are transferred to a tetrazolium redox dye which acts as a terminal electron acceptor turning purple during the reduction [31]. This assay has recently emerged as a simple and reproducible tool for the assessment of mitochondrial metabolism thereby advancing our knowledge of various diseases and pathologies [32-36].

We have added a data interpretation section (Section 4.6) into the methods on line 485.

‘After the 6 hour time period the final OD590 reading was subtracted against the initial reading to calculate the change in dye reduction from electrons flowing into and through the electron transport chain, this value is presented as peak metabolic output. The OmniLog data analysis 1.7 software calculates the maximal rate change from the gradient of the OD590 output during the 6 hour time period. These data are either presented as raw OD590 (A.U.) or as a relative fold change to control conditions.’

The data in Figure 1b and Figure 4d had been plotted as ‘metabolic output (% contribution)’. The legends of these figures had described that these had been calculated as

‘the maximum metabolic peak is averaged for each of the glycolysis, TCA cycle, amino acid, and fatty acid substrate groups and plotted as a percentage of total metabolic output’

However, as the technique does not provide a choice of substrate, we agree that the data presented as ‘percentage contribution’ may be misleading as this was simply the capacity to contribute this percentage and not the actual percentage contribution in competition with other substrates. As such, the data in Figure 1b and Figure 4d has been replotted as an ‘average metabolic output’ for each substrate group. The figure legends have been updated and the text that refers to this data has been changed on…

 Line 125

‘In both the pluripotent D0 stage and the cardiomyocyte D30 stage, TCA substrate metabolism accounted for the greatest average metabolic output, contributing 104.9 A.U. and 220.6 A.U. respectively (Figure 1b).’      

Line 130

‘The greatest increase in average metabolic output was seen in fatty acid substrate metabolism, increasing from 50.5 A.U. at D0 to 137.2 A.U. at D30.’

Line 214

‘When averaged, metabolic output of glycolytic substrates had the smallest reduction, only decreasing from 130.8 A.U. to 113.0 A.U. in ID (Figure 4d).’

Line 316

‘Here, we showed the greatest increase in average metabolic output, and greatest change in metabolism rate, was in fatty acid metabolism following the differentiation of hiPSC-CMs (Figure 1-2).’

  • A bit more background to the role of iron in metabolism would be helpful, such as the significance of iron-sulphur clusters in the electron transport chain.

A paragraph has been added to the introduction that introduces iron metabolism and the importance of iron-sulphur clusters on line 71.

‘Iron sulphur (FeS) clusters are ancient and highly preserved structures that are versatile enzymatic co-factors in numerous metabolic pathways [21]. For example, the mammalian complex I (NADH: ubiquinone oxidoreductase) of the electron transfer chain, one of the largest and well-characterized protein complexes in the cell [22], contains eight essential FeS clusters. Complex I acts as the entry-point of electrons into the electron transfer chain (ETC) and is therefore a key enzyme for aerobic metabolism. Mechanistically, after NADH oxidation by the flavin mononucleotide site, electrons are transferred through a chain of seven FeS clusters and used to reduce ubiquinone to ubiquinol at the inner mitochondrial membrane. Loss of FeS clusters due to defective biogenesis or iron deficiency causes metabolic reprogramming with citrate accumulation and cytosolic lipid drop formation [23]. In the heart, loss of NDUSF4, an FeS cluster containing subunit of complex I, causes reduced complex I activity, cardiac dysfunction and left ventricular hypertrophy [24]. However, the exact metabolic changes that occur in cardiomyocytes during iron deficiency have not been elucidated yet.’

  • In Figure 1 it is clear from the graphs in 1D that the drop in max metabolic rate is small compared to the increase at day 20 – this isn't clear from the text.

We agree the reduction at D6 and D10 is relatively small (averaging a 30.5% reduction). We have added this information into the text on line 122.

‘At the early stages of hiPSC-CM differentiation (D6 and D10) the maximal metabolic level of all substrates was decreased (averaging a 30.5% reduction) compared to undifferentiated D0 hiPSCs (Figure 1a).’

  • In Fig 1Cii relative expression of various mitochondrial proteins are presented. Does the total expression change - I would have expected mitochondrial number to increase and so total expression to increase?

Previous studies have reported that mitochondrial proteins increase during the differentiation of stem cells into cardiomyocytes (Tamai et al. 2011; Kerscher et al. 2015). Live staining of undifferentiated hPSCs, early hPSC-CMs, and late hPSC-CMs with MitoTracker Green and TMRM has allowed this mitochondrial increase to be visualised (Dai et al. 2017).

In our present study we presented the data as the relative abundance of the protein complexes at each timepoint, as we were investigating whether there was an increase reliance on specific complexes during different developmental stages or disease states.

As we state in line 131,

‘Despite the increased fatty acid substrate metabolism observed at D30, there were no significant changes in the relative abundance of the mitochondrial electron transfer chain (ETC) protein complexes (Figure 1c)’.    

And in line 184

‘The decreased mitochondrial function observed in iron deficient hiPSC-CMs did not correspond to a significant change in the relative protein expression of any of the mitochondrial ETC protein complexes (Figure 3e).’

Although we had hypothesized significant changes in complex abundance, we believe reporting these findings will still be of interest to the community as this suggests that the metabolic changes observed were not due to gross misalignment of relative complex expression.

Tamai M., Yamashita A., Tagawa Y.-I. Mitochondrial development of the in vitro hepatic organogenesis model with simultaneous cardiac mesoderm differentiation from murine induced pluripotent stem cells. Journal of Bioscience and Bioengineering. 2011;112(5):495–500.

Kerscher P., Bussie B. S., Desimone K. M., Dunn D. A., Lipke E. A. Characterization of mitochondrial populations during stem cell differentiation. Methods in Molecular Biology. 2015;1264:453–463.

Dao-Fu Dai, Maria Elena Danoviz, Brian Wiczer, Michael A. Laflamme, Rong Tian. Mitochondrial Maturation in Human Pluripotent Stem Cell Derived Cardiomyocytes. Stem Cells Int. 2017; 5153625.

  • Line 117: You say ‘In both D0 hiPSCs 117 and D30 hiPSC-CMs, the highest rates of metabolism were observed with direct TCA cycle substrates, with rates of 16.9 A.U. and 75.0 A.U., respectively’ Is this statistically significant, no stats are given between groups? Also, what does this tell us, since the TCA cycle is used in metabolism of glucose, amino acids and fatty acids? 

The reviewer correctly points out that the highest rates observed in TCA cycle substrates was not significant and added little insight into the data. As such, we have removed the sentence from the text. 

  • Figure 2: the change seen in ‘no substrate’ is interesting. Presumably the cells are in substrate-free media and so are metabolising stored substrates such as glycogen and lipids? They must be using something since there is a change as the cells differentiate.

This is an interesting point and something we had discussed ourselves but had failed to include in the text. We have now added this into the discussion on line 291, highlighting this observation.

‘Indeed, as the cells differentiated their metabolic output increased even when assayed in media devoid of any substrate. This may indicate that the differentiated cardiomyocytes have increased energy reserves, metabolising stored substrates such as glycogen and lipids.’

  • Line 140 add in number of days of transferrin rescue to main text as well as having it in the methods.

This has been added to the text (now line 163).

  • Line 154 What substrates were available in the Seahorse experiment – were there fatty acids in the media or just glucose?

The mitochondrial Seahorse assay was performed as per manufacturer’s instructions. The Seahorse XF Media was supplemented with 1 mM Sodium pyruvate, 2 mM L-Glutamine, and 10 mM Glucose. This information has been added to the methods section of the paper on line 468.

  • Line 208 You say ‘Our results further uncovered which substrates are dysregulated upon iron deficiency.’ This should refer to pathways rather than substrates or say 'Our results further uncovered which substrates have dysregulated metabolism upon iron deficiency.'

This has been changed in the text (now line 221).

  • Line 276 You should mention other oxygen consumption assays, such as the Clarke electrode or Oroborus, which do allow substrate specific oxygen consumption measurements to be made, and radiolabelled experiments which allow substrate utilisation to be measured. All these give different information that can be combined to give a more complete picture, to which the mitochondrial measurements made here would provide valuable additional insights.

Discussion on these techniques has been added into the discussion starting line 274.

‘Mitochondrial function may also be assessed through use of a Clarke’s electrode or Oroborus, both of which allowing individual substrates or inhibitors to be tested for their effect on oxygen consumption. Similarly, radiolabelling of specific substrates can be used to interrogate specific metabolic pathways. Although these methods can provide valuable insights into substrate specific metabolism of cells, they are time-consuming and expensive and may therefore not be appropriate for the simultaneous study or screening of numerous modulators of metabolic pathways.’

  • In Line 302 you don’t discuss why aconitase activity might increase with differentiation. This could be due to the switch from aerobic glycolysis in iPSCs and early differentiation, where glycolytic substrates exit the TCA cycle at citrate for de novo lipogenesis, to oxidative metabolism with full use of the TCA cycle.

This discussion point has now been added into the text on line 303.

‘This increased activity may in part be explained by the switch from aerobic glycolysis, supplying citrate for various anabolic processes including lipogenesis, towards oxidative metabolism through the TCA cycle.’

  • Thankyou for figure 6 – it is very clear and helpful.

Many thanks- we are pleased that this figure was found to be useful
